# Molecular basis for disassembly of an importin: ribosomal protein complex by the escortin Tsr2

Sabina Schütz[1], Erich Michel [2], Fred F. Damberger [3], Michaela Oplová[1,4], Cohue Peña[1], Alexander Leitner [5], Ruedi Aebersold [5,6], Frederic H.-T. Allain[3] & Vikram Govind Panse[1]

Disordered extensions at the termini and short internal insertions distinguish eukaryotic ribosomal proteins (r-proteins) from their anucleated archaeal counterparts. Here, we report an NMR structure of such a eukaryotic-specific segment (ESS) in the r-protein eS26 in complex with the escortin Tsr2. The structure reveals how ESS attracts Tsr2 specifically to importin:eS26 complexes entering the nucleus in order to trigger non-canonical RanGTP-independent disassembly. Tsr2 then sequesters the released eS26 and prevents rebinding to the importin, providing an alternative allosteric mechanism to terminate the process of nuclear import. Notably, a Diamond–Blackfan anemia-associated Tsr2 mutant protein is impaired in binding to ESS, unveiling a critical role for this interaction in human hemato-poiesis. We propose that eS26-ESS and Tsr2 are components of a nuclear sorting system that co-evolved with the emergence of the nucleocytoplasmic barrier and transport carriers.

[1] Institute of Medical Microbiology, University of Zurich, 8006 Zurich, Switzerland. [2] Department of Biochemistry, University of Zurich, 8057 Zurich, Switzerland. [3] Institute of Molecular Biology & Biophysics, ETH Zurich, 8093 Zurich, Switzerland. [4] Institute of Biochemistry, ETH Zurich, 8093 Zurich, Switzerland. [5] Institute of Molecular Systems Biology, ETH Zurich, 8093 Zurich, Switzerland. [6] Faculty of Science, University of Zurich, Zurich 8057, Switzerland. These authors contributed equally: Sabina Schütz, Erich Michel. Correspondence and requests for materials should be addressed to F.H.-T.A. (email: allain@mol.biol.ethz.ch) or to V.G.P. (email: vpanse@imm.uzh.ch)

Membrane-enclosed organelles are a hallmark of eukaryotic cells. This complexity necessitates dedicated machineries that target cargos to their correct compartments, and then ensure their transport to their functional destination.

Components of the replication, transcription, DNA-repair, and ribosome assembly machineries need to be imported into the nucleus. Conversely, messenger ribonucleoproteins and pre-ribosomes assembled in the nucleus are exported into the cytoplasm. Transport receptors (importins and exportins) bind these cargos via signal sequences and translocate them through nuclear pore complexes, embedded within the nuclear envelope. The GTPase Ran regulates interaction of cargos with transport factors, providing directionality to the transport process.[1] In the nucleus, interaction of the importin:cargo complex with RanGTP induces cargo release and terminates the process of nuclear import.[2] While the principles that target cargos to the nuclear compartment are established, how cargos reach their final destination after being released from their importins is unclear.

Eukaryotic ribosome assembly is coordinated by >200 factors that mature pre-ribosomal RNA and simultaneously incorporate ribosomal proteins (r-proteins).[3,4] This conserved process accounts for a major proportion of the nucleocytoplasmic transport in the model organism budding yeast.[5–9] During a single generation time of 90 min a yeast nucleus imports ~14 million r-proteins and at the same time exports 200,000 pre-ribosomes into the cytoplasm. Thus, efficient transport of r-proteins to assembling pre-ribosomes is essential during ribosome production. r-proteins are highly abundant proteins that contain disordered elements making them prone to aggregation and degradation in their non-assembled state.[10] Transport of r-proteins and subsequent transfer to their rRNA-binding site is a formidable challenge. Altered stoichiometry of r-proteins is the underlying cause of Diamond–Blackfan anemia (DBA), a congenital red blood cell aplasia.[11–14] Given the importance and high demand for r-proteins during ribosome formation, specialized mechanisms have evolved to target and protect them during their journey to their assembly site. A set of dedicated chaperones (Syo1, Rrb1, Acl4, and Yar1) have been described that capture specific r-proteins (uL18/uL5, uL11, uL3, uL4, and uS3, respectively) in the cytoplasm, and recruit importins to facilitate their transfer to assembling pre-ribosomes.[15–19] In contrast, the r-protein eS26 employs a different mechanism to reach the 90S pre-ribosome. Although eS26, like other r-proteins, employs multiple redundant importins to reach the nucleus, RanGTP does not efficiently release eS26 from importins in vitro. Instead, a nuclear localized unloading factor Tsr2, which we termed escortin, extracts eS26 from the importins without the assistance of RanGTP, and ensures its safe transfer to the pre-ribosome.[9] Recently, another nuclear localized factor, Bcp1, was shown to function as an escortin for the r-protein uL14 (Rpl23) by extracting uL14 from its importins in a RanGTP-dependent manner.[20] Yet, how Tsr2 recognizes an importin:eS26 complex among diverse importin:cargo complexes entering the nucleus remains unclear.

In this study, we reveal that eukaryotic eS26 has acquired segments, over its anucleated archaeal counterparts, which enables Tsr2 to specifically disassemble an importin:eS26 complex. We propose that these segments and the cognate escortin Tsr2 have co-evolved with the emergence of the nuclear envelope and transport receptors.

## Results

### Tsr2 binds eS26 through eukaryotic-specific segments (ESSs).
To reveal the molecular contacts between Tsr2 and eS26, we subjected a recombinant Tsr2:eS26 complex to chemical cross-linking followed by mass spectrometry (XL-MS). Tsr2 and eS26 are challenging for XL-MS studies, since both proteins have an uneven distribution of the Lys, Asp, and Glu residues targeted for chemical crosslinking and proteolysis. Therefore, in addition to the lysine-reactive disuccinimidyl suberate (DSS), we subjected the Tsr2:eS26 complex to a combination of adipic or pimelic acid dihydrazide (ADH and PDH, respectively) and 4-(4,6-dimethoxy-1,3,5-triazin-2-yl)-4-methylmorpholinium chloride (DMTMM) that results in crosslink products between two Asp or Glu residues incorporating the dihydrazide, as well as zero-length crosslinks between Lys and Asp or Glu residues.[21] The crosslinked complexes were digested with proteases trypsin and Glu-C, and the products were fractionated by size-exclusion chromatography and analyzed by liquid chromatography-tandem mass spectrometry.[22] By combining different crosslinking chemistries, we identified 30 crosslinked peptides (14 on eS26, 4 on Tsr2, and 12 connecting the two proteins), corresponding to 24 non-redundant residue-residue contacts (Supplementary Table 1). Crosslinks within Tsr2 could only be observed with Glu-C as the protease, while all intra-eS26 links were identified after trypsin digestion. Inter-protein crosslinks between eS26 and Tsr2 were all tryptic peptides, but all crosslinking chemistries contributed complementary and confirmatory data to the connectivity map[23] (Fig. 1a). Residues, especially in the C-terminal region of eS26, crosslinked with residues along the N-terminal domain of Tsr2 (1–152), hereafter termed Tsr2-N.

eS26 is absent in bacteria, but present in most archaea (Fig. 1b). Eukaryotic eS26 is characterized by a short five amino acid insertion (68–72), eukaryotic-specific segment 1 (ESS1), and a 20-amino acid C-terminal extension (99–119), eukaryotic-specific segment 2 (ESS2; Fig. 1c). While ESS1 forms part of a β-sheet, ESS2 was not resolved in the crystal structure of the 40S subunit.[24] In light of the XL-MS data, and that Tsr2 is present only in eukaryotes, we wondered whether Tsr2 binds eS26 through ESS1 and ESS2. We incubated immobilized GST-Tsr2 with different variants of recombinant eS26 wherein either both ESSs (eS26$^{ESS1+2\Delta}$) were removed or one of the ESSs (eS26$^{ESS1\Delta}$ and eS26$^{ESS2\Delta}$) was deleted. The assays were performed in PBS buffer (low salt) and in PBS buffer supplemented with 500 mM NaCl (high salt). eS26 lacking either ESS1 or ESS2 bound to GST-Tsr2 at low salt (Fig. 1d, lanes 1–3). eS26 deficient in both ESSs failed to bind Tsr2 in these conditions (Fig. 1d, lane 4). In high-salt conditions, WT eS26 and the eS26 variant lacking ESS1 efficiently bind to GST-Tsr2 (Fig. 1d, lanes 6–7). An eS26 variant lacking ESS2 did not bind to GST-Tsr2 under these conditions (Fig. 1d, lane 8). Removal of a short five amino acid segment within ESS2 (Δ99–104) strongly impaired interactions between GST-Tsr2 and the eS26 variant at high-salt conditions (Fig. 1e, lane 6).

We investigated the phenotypes associated with ESS-deficient eS26 variants. To this end, we transformed plasmids encoding WT eS26, eS26$^{ESS1\Delta}$, eS26$^{ESS2\Delta}$ and eS26$^{ESS1+2\Delta}$ into the conditional $P_{GAL1}$-RPS26Arps26bΔ strain and analyzed the growth of the resultant strains on repressive glucose containing media (Fig. 1f). The eS26$^{ESS1+2\Delta}$ mutant did not complement the lethality of the $P_{GAL1}$-RPS26Arps26bΔ conditional mutant. Although eS26$^{ESS1\Delta}$ and eS26$^{ESS2\Delta}$ rescued the lethality of the $P_{GAL1}$-RPS26Arps26bΔ strain, both mutants showed severe growth defects (Fig. 1f). Fluorescence in situ hybridization using a Cy3-labeled probe against the internally transcribed spacer sequence 1 (ITS1) downstream of 18S rRNA revealed that both mutants accumulated 20S pre-rRNA in the cytoplasm indicating defects in ITS1 cleavage (Fig. 1g). Consistent with interaction studies (Fig. 1e, lane 6), a $P_{GAL1}$-RPS26Arps26bΔ strain expressing eS26 lacking residues

99–104 within ESS2 showed strongly impaired growth on repressive glucose containing media (Fig. 1f).

Overexpressing a ProtA-FLAG-eS26 fusion that cannot be incorporated into the 40S subunit[9] impaired growth of yeast cells at lower temperatures in a dominant-negative manner. In contrast, overexpressing a construct lacking ESS2 (ProtA-FLAG-eS26ΔESS2) did not show this phenotype (Fig. 1h). Pull-down experiments indicate that ProtA-FLAG-ESS2 co-enriched Tsr2 demonstrating that ESS2 alone can titrate Tsr2 away from eS26 in vivo (Fig. 1i).

Altogether, these data support the conclusion that both ESSs contribute to interaction between Tsr2 and eS26.

**NMR structure of Tsr2-N.** To reveal how Tsr2 recognizes eS26, we analyzed recombinant Tsr2 and the Tsr2:eS26 complex by NMR spectroscopy. XL-MS studies revealed that eS26 crosslinks with Tsr2-N (Fig. 1a). Two-hybrid and binding assays indicated that Tsr2-N is sufficient to interact with eS26 (Supplementary Fig. 1a and b, lane 6). A $P_{GAL1}$-TSR2 conditional yeast mutant-expressing Tsr2-N grew at nearly WT rates (Supplementary

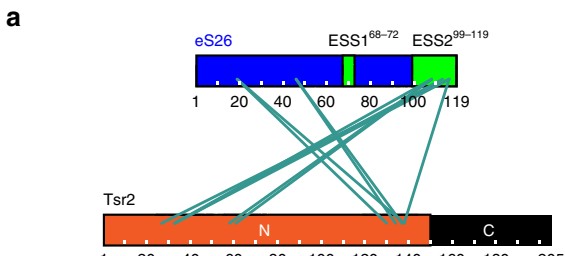

Fig. 1c) and was not impaired in 20S pre-RNA processing (Supplementary Fig. 1d), indicating that the Tsr2-N is sufficient to deliver eS26 to the pre-ribosome. Therefore, we focused on Tsr2-N for structural studies.

A 2D $^{15}$N,$^1$H-HSQC spectrum of $^{15}$N-labeled Tsr2-N showed well-dispersed resonances, indicative of a folded protein (Supplementary Fig. 2a). NOE assignment provided 3818 interproton distance constraints as input for the structure calculation, which was further complemented with 521 backbone and sidechain torsion angle constraints. Refinement in implicit water using AMBER12 yielded a well-defined structure of Tsr2-N with an rmsd of 0.50 and 0.89 Å for the backbone and heavy atoms of the structured region of the protein (residues 10–140), respectively (Fig. 2a, Supplementary Table 2).

The Tsr2-N structure consists of six tightly packed α-helices comprised of residues 28–42 (α1), 44–52 (α2), 59–74 (α3), 82–95 (α4), 106–121 (α5), and 126–139 (α6) and a small anti-parallel β-sheet of residues 11–13 (β1) and 79–81 (β2) (Fig. 2b, PDB-6G03). The β-sheet was confirmed by manual inspection of the NOE network at the β-strand interfaces (Supplementary Fig. 2b). The $\{^1$H$\}^{15}$N steady-state heteronuclear NOE experiment, which is a measure of the local sub-nanosecond mobility of individual amide bonds, revealed that the compact fold of Tsr2-N is well-ordered, and only residues at the termini and in the loops connecting β1 to α1 and α4 to α5 show conformational mobility (Fig. 2a, Supplementary Fig. 2c). A search for structural homologs to Tsr2-N using the DALI protein structure comparison web server (http://ekhidna2.biocenter.helsinki.fi/dali/) indicated a modest similarity (Z-score of 7.5 and an rmsd of 2.7 Å for only five α-helices excluding α2 and the β-sheet) only to the N-terminal domain of the mRNA export factor Nab2 (PDB-2JPS) (Supplementary Fig. 2d).

To obtain insights into the acidic C-terminal tail of Tsr2 (Tsr2-C$^{153–205}$), we sought to assign the backbone resonances of full-length Tsr2 in its free form (Supplementary Fig. 2e). Despite strong signal degeneracy, we assigned backbone resonances for all residues except for E169, D170, M173, E175, and R203. No secondary structure was detected in the conformationally mobile acidic tail, and this was supported by $\{^1$H$\}^{15}$N NOE data (Fig. 2c).

**NMR structure of Tsr2-N in complex with ESS2**. We aimed to determine the structure of the Tsr2:eS26 complex by isotope labeling each component. As eS26 is aggregation prone, we were only able to prepare the $^{13}$C,$^{15}$N-labeled Tsr2:eS26 complex by co-expressing both components (Supplementary Fig. 3a). We assessed both the local mobility and chemical shift perturbation

of Tsr2 upon binding eS26 (Fig. 2c, d). Chemical shift perturbations were distributed throughout Tsr2 with a conspicuous cluster of affected residues in helix α3, the C-terminal part of helix α4, and the loop between helices α4 and α5 (Fig. 2e). The large chemical shift perturbations at the C-terminal end of α6 and the beginning of the acidic tail suggest that this region contributes to the interaction with eS26, and this conclusion is further corroborated by $\{^1$H$\}^{15}$N NOE data that show an increase in conformational rigidity (Fig. 2c). Even in complex with eS26, the beginning of the acidic tail of Tsr2 (160–175) remained significantly mobile than the folded Tsr2-N domain, but less mobile than in the free state indicative of loose dynamic interactions between the acidic tail of Tsr2 and eS26.

To obtain further insights into the interactions between eS26 and Tsr2, we assigned the resonances of eS26 bound to Tsr2. However, significant signal overlap allowed only assignment of residues 5–8, 20–37, 40–44, 46–51, 65–76 (which includes ESS1), and 101–119 of eS26 (which includes ESS2, Supplementary Fig. 3a, b). $C_\alpha$ and $C_\beta$ chemical shifts support a similar secondary structure to eS26 on the mature 40S (Supplementary Fig. 3b). ESS2 which is not visible in the 40S crystal structure (Supplementary Fig. 3c) adopts an α-helical structure in the Tsr2:eS26 complex. $^{15}$N$\{^1$H$\}$ NOE experiments indicated that the assigned eS26 backbone amides were rigid, including all residues comprising ESS2 (Supplementary Fig. 3b).

We characterized the interaction of Tsr2-N with a ESS2 peptide, and solved the NMR structure of the Tsr2-N:eS26-ESS2 complex. Tsr2 binds to ESS2 in vitro with a $K_d$ of 0.7 μM at 1:1 stoichiometry (Supplementary Fig. 4a). NMR titration of Tsr2-N with the ESS2 peptide showed slow exchange behavior for signals of both components on the NMR timescale. Resonance assignments of the protein–peptide complex are nearly complete (Supplementary Fig. 4b, c). The final structure was calculated using 4418 NOE-derived distance constraints including 331 intermolecular constraints and a list of 28 manually defined upper distance limits between Tsr2-N and ESS2 as well as 601 backbone and sidechain torsion angle constraints. We obtained a well-defined structure of the Tsr2-N:ESS2 complex with an rmsd of 0.46 and 0.97 Å for the backbone and heavy atoms of the structured regions (residues 10–140 of Tsr2-N and residues 100–119 of eS26), respectively (Supplementary Table 3).

No major conformational changes were observed in Tsr2-N upon ESS2 binding (rmsd of 1.11 Å between the average backbone coordinates of residues 10–140 from the two structures, Fig. 3a, PDB-6G04). In contrast, ESS2 undergoes a transition from a disordered peptide as indicated by $^{15}$N$\{^1$H$\}$ NOEs and

**Fig. 1** Eukaryotic-specific segments of eS26 are required to bind Tsr2. **a** XL-MS reveals crosslinks between ESS2 and N-terminal domain of Tsr2. The crosslinked residues are listed in the Supplementary Table 1. **b** Phylogenetic analyses for eS26 and Tsr2. ESS1, ESS2 from eS26 and Tsr2 are present only in eukaryotes. **c** Sequence alignment of yeast S26 compared to the indicated species.[70] **d** ESSs in eS26 are required to bind Tsr2 in vitro. GST-Tsr2 was immobilized on Glutathione Sepharose before incubation with *E. coli* lysate containing recombinant WT eS26, eS26 deficient in ESS1 and/or ESS2 or archaeal eS26 from *Sulfolobus solfataricus*. Bound proteins were eluted by SDS sample buffer, separated by SDS-PAGE and visualized by Coomassie Blue staining. L = input (1:10 diluted). **e** Residues 99–109 in eS26-ESS2 are necessary to bind Tsr2 in vitro. GST-Tsr2 was immobilized on Glutathione Sepharose before incubation with an *E. coli* lysate containing recombinant WT eS26 or eS26 with variant truncations in C-terminal ESS2. Samples were analyzed as in **d**. Results from in vitro binding were quantified using ImageJ. **f** ESS1 and ESS2 deletion from eS26 causes slow growth phenotype in yeast. The conditional P$_{GAL1}$-RPS26Arps26bΔ strain was transformed with WT or the indicated truncations of eS26 and spotted in 10-fold dilutions on repressive glucose containing media and grown at 25 °C for 4 days. **g** Cells with eS26 lacking ESS1 or ESS2 accumulate immature 20S pre-rRNA in the cytoplasm. Localization of 20S pre-rRNA in P$_{GAL1}$-RPS26Arps26bΔ cells transformed with indicated plasmids was analyzed by FISH using a Cy3-labeled oligonucleotide complementary to the 5′ portion of ITS1 (red). Nuclear and mitochondrial DNA was stained with DAPI (blue). Scale bar = 5 μm. **h** Overexpression of ProtA-FLAG-eS26 is toxic in yeast. The WT yeast strain (BY4741) was transformed with ProtA-FLAG-eS26 or ProtA-FLAG-eS26 lacking ESS2, spotted in 10-fold dilutions on galactose containing media and grown at 25 °C for 4 days. **i** FLAG-ESS2 fusion protein co-precipitates Tsr2. ESS2 was purified using ProteinA-Tev-FLAG tag, the FLAG eluate was TCA precipitated, separated by SDS-PAGE, and analyzed by Coomassie staining and western analyses using the indicated antibodies

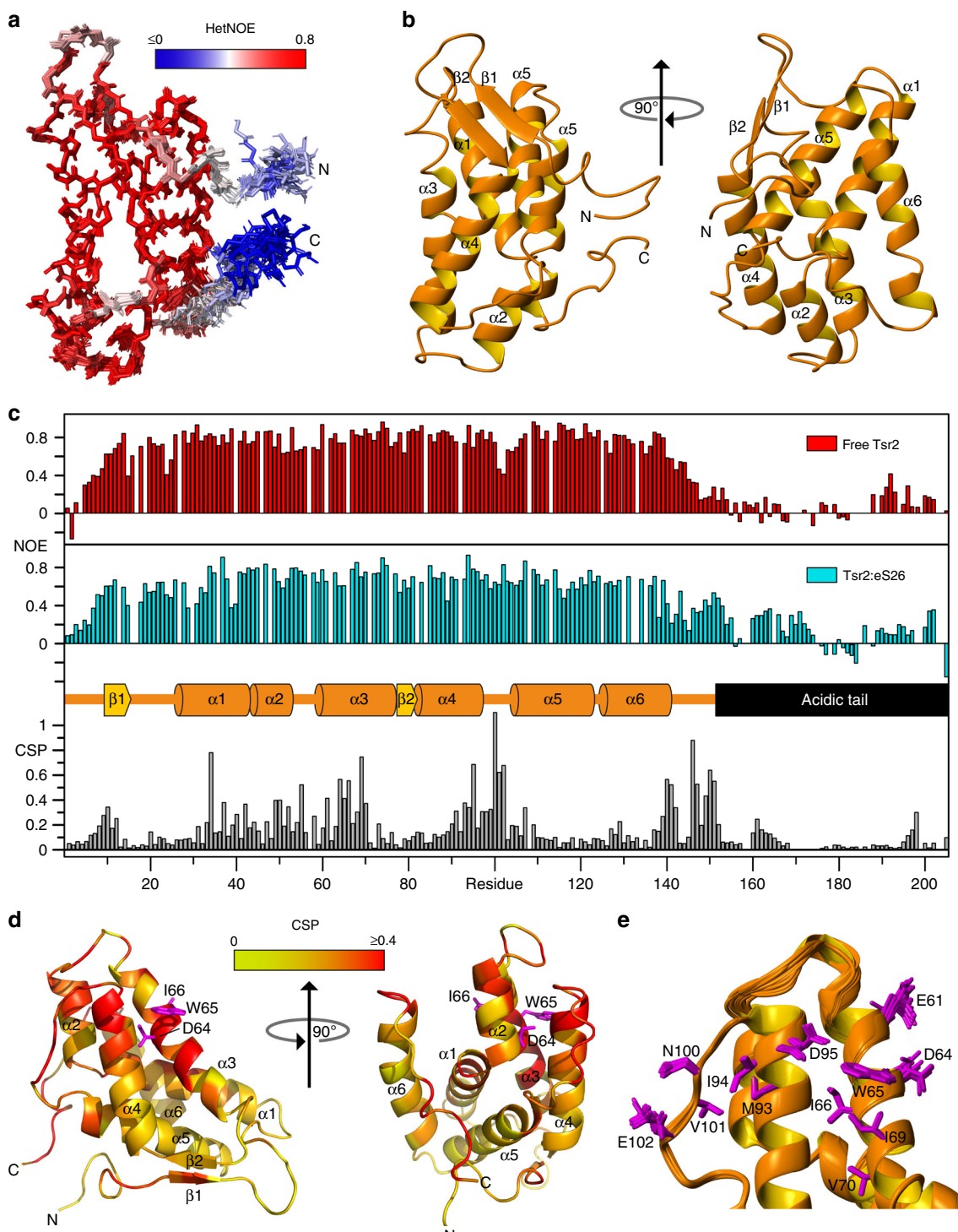

**Fig. 2** Structure of Tsr2-N and in complex with eS26 revealed by NMR. **a** Superposition of the 20 lowest-energy conformers representing the NMR solution structure of Tsr2-N after energy-minimization with AMBER. The N- and C-termini are indicated with N and C, respectively. The color coding reflects the conformational mobility of individual backbone amides derived from the {$^1$H}$^{15}$N NOE experiment and ranges from blue to red for flexible and rigid moieties, respectively. **b** Cartoon representation of Tsr2-N with labels indicating α-helices and β-strands. **c** Upper panel: {$^1$H}$^{15}$N NOE data of full-length Tsr2 in the absence and presence eS26. Lower panel: The combined $^1$H and $^{15}$N chemical shift perturbation (CSP) of full-length Tsr2 upon binding to eS26. Secondary structure boundaries of Tsr2-N as well as the acidic tail region are indicated in the center. **d** CSP data from **c** visualized on a cartoon representation of the lowest-energy conformer of Tsr2-N. The color coding reflects the perturbation of individual amide resonances and ranges from yellow (CSP = 0 ppm) to red (CSP ≥ 0.4 ppm). Sidechains of residues D64, W65, and I66 are shown. **e** Sidechains of residues with a particularly large CSP are indicated in magenta on a cartoon representation of the ensemble of 20 Tsr2-N conformers

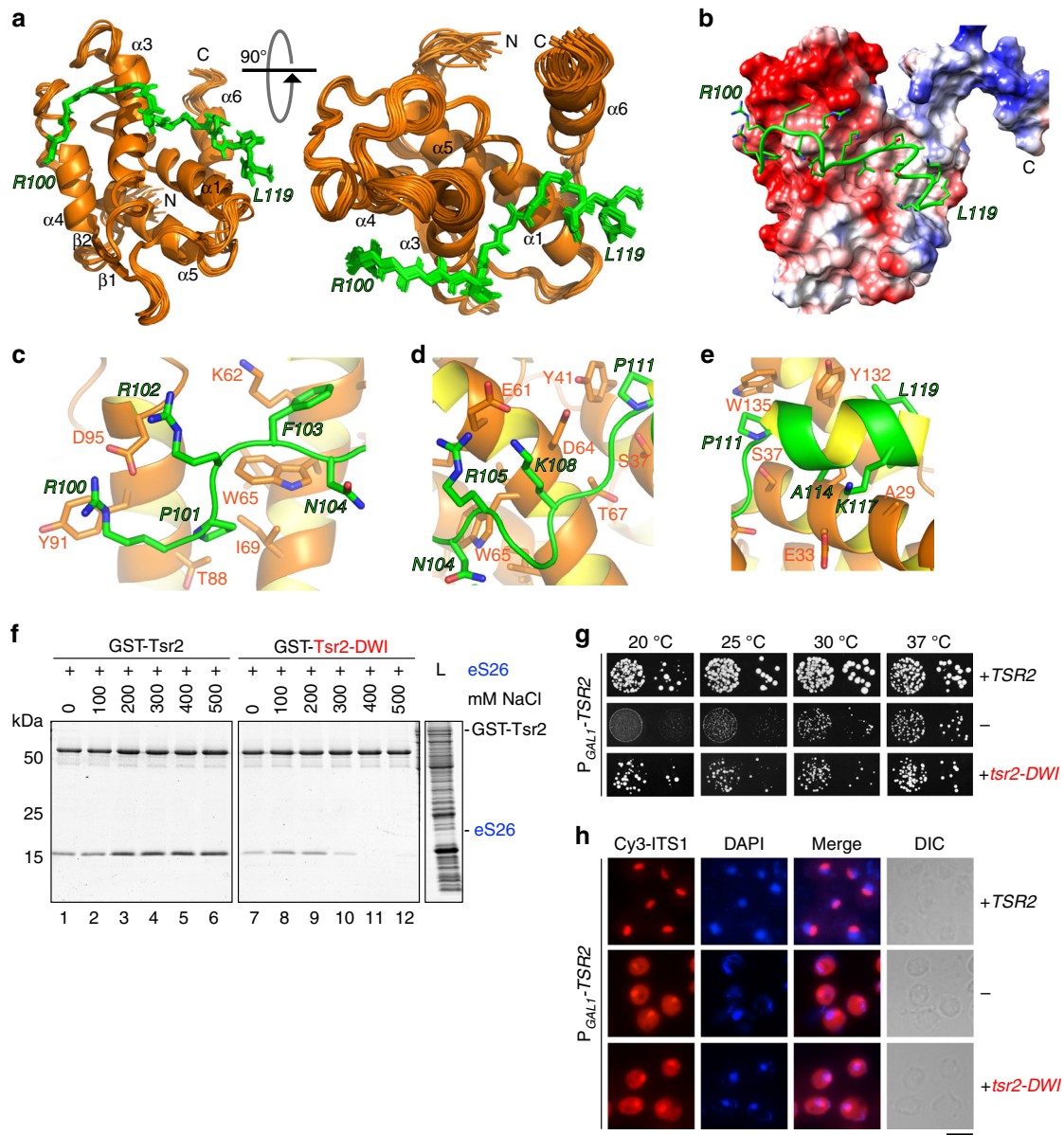

**Fig. 3** Molecular interactions between Tsr2 and ESS2. **a** Superposition of the 20 lowest-energy conformers representing the solution structure of Tsr2-N in complex with ESS2. **b** Coulomb potential projected onto the solvent accessible surface of Tsr2 with the backbone of ESS2 represented by tube; sidechains are presented by sticks with oxygens red, carbons green, nitrogens blue, and hydrogens white. The potential due to Tsr2 is represented by red for negative values (acidic residues) and blue for positive values (basic residues). The C-terminus of Tsr2-N is visible as the blue arm extending to the right. **c–e** Closeup of various interactions between residues of TSR2-N (orange letters label sidechains shown in stick representation) and ESS2 (green italic letters). Sidechain atoms are colored as in **b**. **f** Tsr2[DWI] mutant shows impaired interaction with eS26 in vitro. GST-Tsr2 or GST-Tsr2[DWI] was incubated with recombinant eS26 at the indicated salt concentrations before pull-down. L = input (1:10 diluted). Results from in vitro binding were quantified using ImageJ. **g** Tsr2[DWI] mutant impairs yeast growth. The conditional P$_{GAL1}$-TSR2 strain was transformed with empty vector or vector with *TSR2 WT* or *tsr2[DWI]*. Transformants were spotted in 10-fold dilutions on repressive glucose containing media and grown at indicated temperatures for 2–4 days. **h** Tsr2[DWI] mutant cells accumulate immature 20S pre-rRNA in the cytoplasm. P$_{GAL1}$-TSR2 cells transformed with WT *TSR2* or *tsr2[DWI]* were grown at 30 °C in glucose containing media to the mid-log phase. Localization of 20S pre-rRNA was analyzed by FISH. Scale bar = 5 μm

random-coil ¹³Cα shifts to a structured conformation in the complex as evidenced by ¹⁵N{¹H} NOEs (Supplementary Fig. 4b–d). ESS2 extends across four of the helices on one face of Tsr2-N with its N-terminal half (100–110) taking on an extended backbone conformation interrupted by kinks due to two prolines causing it to wrap around helix α3. The C-terminal part of eS26 ESS2 forms a short α-helix comprised of residues 111–119, which inserts between α1 and α6 of Tsr2-N (Fig. 3b–e). These data are consistent with the observed α-helix within ESS2 for the full-length eS26 bound to Tsr2 (Supplementary Fig. 3b).

ESS2 packs tightly against the surface of Tsr2-N using a dense network of hydrophobic and electrostatic interactions. The C-terminal α-helix of ESS2 is amphipathic and makes hydrophobic contacts to a cluster of aromatics of Tsr2. P111 of ESS2 contacts Y41, Y132 and W135 of Tsr2-N. The extended N-terminal part of ESS2 makes major hydrophobic contacts with P101 interacting with W65 and I69 of Tsr2, but also numerous electrostatic interactions like R100 and R102, contacting D95 and E96 of Tsr2 (Fig. 3b–e). The importance of this latter set of interactions is supported by the severe impairment in binding to

Tsr2 when residues 99–104 within ESS2 are deleted (Fig. 1e, lane 6). ESS2 conformation is stabilized by a number of intra- and intermolecular H-bonds. In addition to a regular set of H-bonds connecting NH and carbonyl groups of the C-terminal helix, ESS2 also makes intramolecular H-bonds from N104 Hδ22 to N107 Nδ2, and E106 HN to N104 Nδ2, which help stabilize the position of the involved Asn sidechains to form a cage around the W65 sidechain of Tsr2-N. Finally, an intermolecular H-bond is formed from V109 HN of ESS2 to the carbonyl of D64 from Tsr2-N, which represents an unsatisfied H-bond acceptor in α2 helix. This irregularity in the α2 helix geometry is facilitated by G68. A hallmark of this complex is the numerous conserved interactions between the positively charged residues of eS26 (R100, R102, R105, and K108) with the negatively charged surface of Tsr2 (E61, D64, E87, D95, E96) as illustrated in Fig. 3b.

The strong chemical shift perturbations on residues D64, W65 and I66 formed a basis to validate relevance of ESS2:eS26 interactions in vivo. Therefore, we constructed a Tsr2 mutant, Tsr2[DWI], with these three residues mutated to alanine for in vitro and in vivo functional studies. In vitro binding studies revealed that in comparison to WT Tsr2, the Tsr2[DWI] mutant protein is strongly impaired in interacting with eS26 (Fig. 3f, lane 1 vs 7). This impairment was aggravated when binding studies were carried out in increasing salt concentrations (Fig. 3f, lanes 2–6 vs 8–12). Consistent with these studies, a Tsr2[DWI] mutant yeast strain shows impaired growth (Fig. 3g). Like the Tsr2-depletion strain,[9] the Tsr2[DWI] mutant is defective in 20S pre-rRNA processing as judged by cytoplasmic accumulation of Cy3-ITS1 (Fig. 3h).

**ESSs mediate importin:eS26 complex disassembly.** We investigated whether the ESSs are required to trigger Tsr2-mediated disassembly of the importin:eS26 complex. WT eS26 and its ESS deletion variants were incubated with the importin Kap123 in the presence of competing *E. coli* lysates, and levels of the bound eS26 were analyzed by western blotting. Both WT eS26 and all ESS variants efficiently bound to the importin (Fig. 4a, lanes 2, 4, 6, 8) indicating that Kap123 does not interact with eS26 through ESSs. We conclude that eS26 employs distinct sites to bind to Kap123 and Tsr2.

We incubated these preformed importin:eS26 complexes with Tsr2 and analyzed the levels of Tsr2-bound eS26 and its ESS deletion variants by western blotting. Tsr2 efficiently dissociated importin:eS26 and importin:eS26[ΔESS1] complexes (Fig. 4a, lanes 3 and 5). In comparison, the importin:eS26[ΔESS2] was less efficiently dissociated by Tsr2 (Fig. 4a, lane 7). Consistent with interactions studies (Fig. 1d, lane 4), Tsr2 was unable to dissociate the importin:eS26[ΔESS1+2] complex (Fig. 4a, lane 9). Conversely, the Tsr2[DWI] mutant that exhibits reduced affinity to eS26 is also unable to efficiently dissociate a Kap123:eS26 complex (Fig. 4b) indicating that Tsr2-mediated disassembly of the importin:eS26 complex in vitro requires both ESSs.

Co-enrichment of both ESS deletion variants with Enp1-TAP was also strongly impaired (Fig. 4c). These phenotypes are due to reduced eS26 mutant protein levels as judged by western analyses of whole-cell lysates (Fig. 4d). Thus, ESS:Tsr2 interactions are critical to dissociate eS26 from the importin and to protect and target eS26 to the pre-ribosome in vivo.

**DBA-linked Tsr2 mutant is impaired in binding ESS2.** Several mutations in human eS26 that result in haploinsufficiency of the r-protein have been linked to DBA[25] including a E64G in human Tsr2 (hTsr2E64G).[26,27] To investigate the molecular basis underlying the pathogenicity of this mutation, we generated yeast strains expressing WT human Tsr2 (hTsr2) and hTsr2E64G and analyzed the phenotypes associated with these strains. Yeast cells expressing hTsr2E64G were strongly growth impaired, (Fig. 5a) and showed defective 20S pre-rRNA as judged by the strong cytoplasmic accumulation of Cy3-ITS1 (Fig. 5b).

To evaluate whether the hTsr2E64G mutant was impaired in interacting with human eS26, we incubated immobilized GST-hTsr2 and GST-hTsr2E64G with human eS26 under a range of NaCl concentrations, and analyzed the levels of the bound r-protein by SDS-PAGE. Binding to human eS26 is weaker for hTsr2E64G mutant than the WT (Fig. 5c). This weaker interaction is not due to perturbed folding of hTsr2E64G since far-UV circular dichroism (CD) measurements revealed identical secondary structure content for mutant and WT proteins (Fig. 5d, upper panels). Furthermore, the melting temperature curves determined by far-UV CD spectroscopy indicated nearly identical transition midpoints of ca. 57.6 °C and ca. 57.4 °C for the WT and mutant Tsr2, respectively (Fig. 5d, lower panels).

Given that, ESS2 makes important contributions to the Tsr2:eS26 interactions, we tested whether the hTsr2E64G was perturbed in its interaction with human ESS2 (hESS2). To this end, we measured the affinity of hESS2 to hTsr2 and hTsr2E64G by isothermal titration calorimetry (ITC) (Fig. 5e). Binding of WT hTsr2 to hESS2 showed a $K_d$ of 2 μM, which is in the range of the value measured for the yeast complex (Supplementary Fig. 4). In contrast, the hTsr2E64G mutant displayed a 45-fold reduced affinity to hESS2 ($K_d$ of 90 μM) confirming that the DBA-associated mutation in hTsr2 has a significantly weakened interaction with human eS26. eS26 overexpression restored slow growth of the human Tsr2E64G expressing strain to nearly WT growth rates (Fig. 5a) supporting the notion that the interactions between hTsr2 and human eS26 are weakened in these DBA patients.

**The C-terminal acidic tail of Tsr2 keeps eS26-RNA free.** During our attempts to purify recombinant eS26 for biochemical studies, we noticed that GST-eS26 co-enriched with RNA, as judged by the high absorbance ratio at 260 versus 280 nm (Fig. 6a). To obtain a RNA free r-protein, we treated the purified GST-eS26 with RNase A. This led to aggregation of GST-eS26 fusion protein (Fig. 6b). Since Tsr2 prevents aggregation of eS26 in vitro,[9] we tested whether Tsr2 is able to suppress this RNAse A-induced aggregation of the GST-eS26 fusion. Supplementing the reaction with Tsr2, but not Tsr2-N, robustly suppressed RNAse A induced aggregation of GST-eS26 as judged by decreased light scattering at 450 nm (Fig. 6c).

We investigated how Tsr2 suppressed this RNase A-induced GST-eS26 aggregation. We treated immobilized GST-eS26 with increasing amounts of Tsr2 or Tsr2-N, and then extracted the RNA bound to GST-eS26. We found that the RNA bound to GST-eS26 (Fig. 6d, lane 3) was released upon incubation with Tsr2, but not Tsr2-N (Fig. 6d, compare lanes 4–5 and 9–10). The C-terminal tail of Tsr2 (Tsr2-C) is enriched in acidic amino acids (Supplementary Fig. 5). We wondered whether this region competes with RNA to bind eS26. We incubated immobilized GST-Tsr2-C with lysates containing WT eS26 and Tsr2-N in the presence of RNase A (Fig. 6e). The bound proteins were separated on an SDS gel and analyzed by Coomassie Blue staining and western blotting. Tsr2-N and eS26 bound to GST-Tsr2-C only when they were both present together with RNase A. In the absence of RNase A, Tsr2-N and eS26 were poorly recruited to GST-Tsr2-C. Incubation with Tsr2 and RNase A did not permit loading of the Tsr2:eS26 complex onto GST-Tsr2-C. These data indicate eS26 binds to Tsr2-C and Tsr2-N through distinct binding sites.

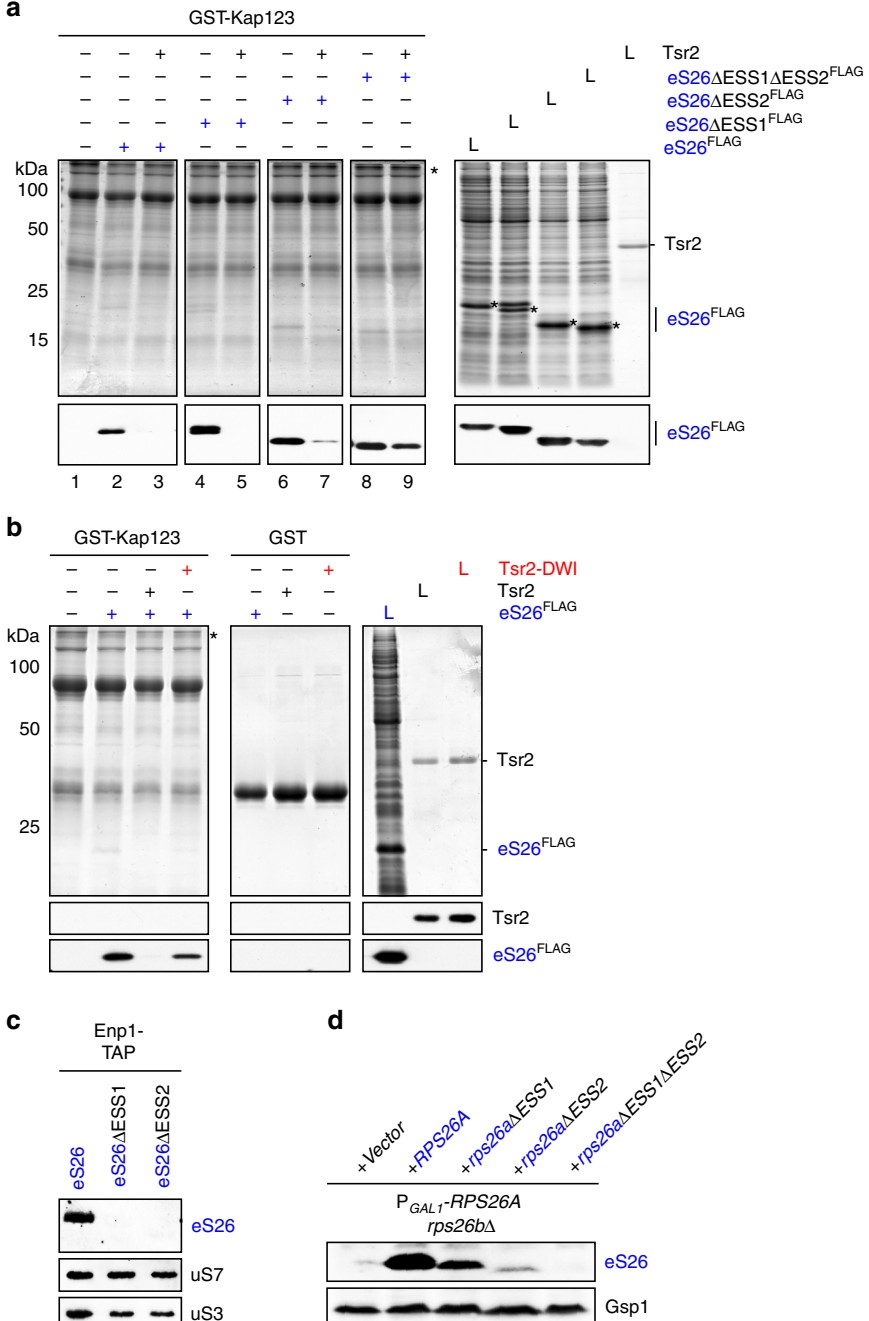

**Fig. 4** ESSs mediate Tsr2-dependent importin:eS26 complex disassembly. **a** Tsr2 cannot efficiently dissociate the Kap123:eS26ΔESS1ΔESS2$^{FLAG}$ complex. The complex of GST-Kap123 with eS26$^{FLAG}$, eS26ΔESS1$^{FLAG}$ eS26ΔESS2$^{FLAG}$ or eS26ΔESS1ΔESS2$^{FLAG}$ was immobilized on Glutathione Sepharose and incubated with either buffer alone or with Tsr2 before pull-down. L = input (1:10 diluted). **b** Tsr2$^{DWI}$ mutant does not dissociate a Kap123:eS26 complex. The complex of GST-Kap123 with eS26$^{FLAG}$ was incubated with either buffer alone, purified Tsr2 or Tsr2$^{DWI}$ before pull-down. L = input (1:10 diluted). **c** Efficient recruitment of eS26 to pre-40S requires both of the ESSs. Enp1-TAP was isolated from P$_{GAL1}$-RPS26Arps26bΔ strain transformed with WT eS26 or eS26 lacking ESS1 or ESS2. After tandem affinity purification, eluates were separated by 4–12% gradient SDS-PAGE and subjected to western analyses using indicated antibodies. Protein levels of uS7, uS3 served as a loading control. **d** Protein levels of eS26, eS26ΔESS1, eS26ΔESS2 and eS26ΔESS1ΔESS2 in whole-cell extracts of P$_{GAL1}$-RPS26Arps26bΔ strain were determined by western analyses using α-eS26 antibodies. Gsp1 protein levels served as a loading control

## Discussion

It is generally accepted that after entering the nucleus, cargos bound to their importins are indiscriminately released upon interaction with RanGTP. Yet, it is unclear how cargos are channeled to their functional site after being released from their importins. Here, we provide insights into how the r-protein eS26 while bound to its importin is sorted out for ribosome assembly.

The escortin Tsr2 extracts eS26 from an importin:eS26 complex in a RanGTP-independent manner.[9] Here, we identify eukaryotic-specific segments, ESSs, within eS26, that serve as beacons to attract Tsr2 to the importin:eS26 complex in order to trigger RanGTP-independent disassembly. Removal of ESSs compromises eS26 stability, and impairs targeting of the r-protein to the pre-ribosome. Thus, ESSs link the nucleocytoplasmic

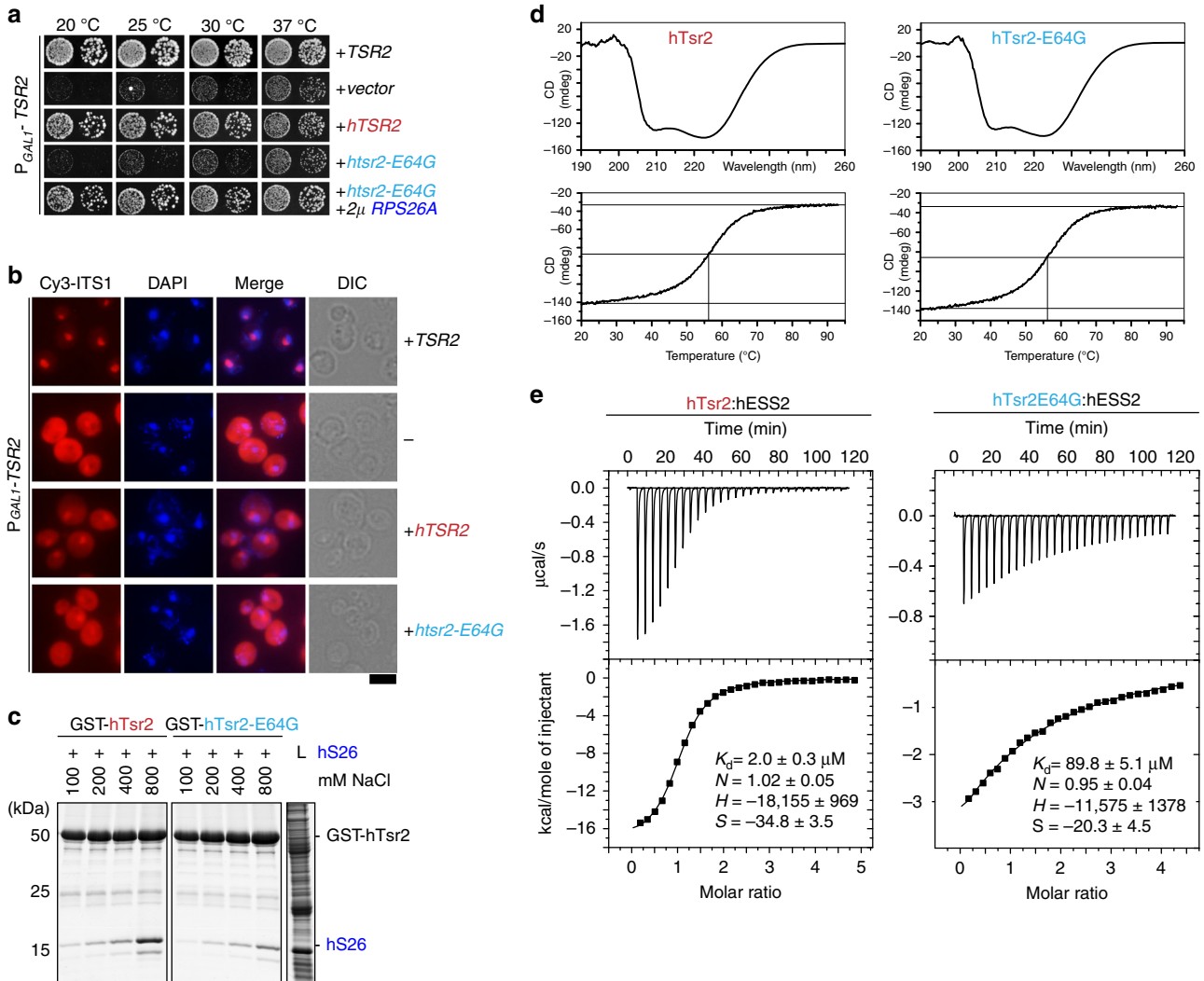

**Fig. 5** DBA-linked Tsr2E64G mutant is impaired in binding ESS2. **a** hTsr2E64G mutant expressed in yeast impairs yeast growth. Overexpression of eS26 rescues from the impaired growth. The conditional P$_{GAL1}$-TSR2 strain was co-transformed with 2μ vectors expressing indicated genes. Transformants were spotted in 10-fold dilutions on repressive glucose containing media and grown at indicated temperatures for 2–4 days. **b** Tsr2 DBA mutant cells (hTsr2E64G) accumulate immature 20S pre-rRNA in the cytoplasm. P$_{GAL1}$-TSR2 cells were grown at 30 °C in glucose containing media to mid-log phase. Localization of 20S pre-rRNA was analyzed by FISH using a Cy3-labeled oligonucleotide complementary to the 5′ portion of ITS1 (red). Nuclear and mitochondrial DNA was stained with DAPI (blue). Scale bar = 5 μm. **c** hTsr2E64G mutant inefficiently binds eS26. GST-hTsr2 and GST-hTsr2E64G were immobilized on Glutathione Sepharose before incubation with *E. coli* lysate containing recombinant eS26 and incubated at indicated salt concentrations. Bound proteins were eluted by SDS sample buffer, separated by SDS-PAGE, visualized by Coomassie Blue staining. L = input (1:10 diluted). The error bars show the standard deviation. **d** The far-UV CD spectra of both human WT Tsr2 and the E64G-DBA variant. The midpoints of the thermal denaturation curves followed at 222 nm are at 57 °C for both proteins. **e** Isothermal titration calorimetry (ITC) measurements of human Tsr2 with human ESS2. The binding isotherms were plotted against the molar ratio. The measured parameters and $K_d$ values are indicated within the plots

transport machinery to ribosome assembly ensuring safe passage of the r-protein to its rRNA-binding site. Forty-six r-proteins contain ESSs over their archaeal counterparts.[28] These segments might serve additional functions including linking the cellular transport machinery with the ribosome assembly pathway.

Dedicated chaperones Syo1, Sqt1, Rrb1, Yar1, and Acl4 were proposed to capture their r-protein clients in the cytoplasm as they emerge from the ribosome via their eukaryotic-specific N-terminal segments.[15,16] uL4 is an exception, wherein a eukaryotic-specific loop provides the binding platform for Acl4 for co-translational capture.[17,29] These chaperones additionally function as adaptors to recruit the nuclear import machinery.[8,30,31] In contrast, eS26 harbors separate sites to bind to its importin and the escortin Tsr2. eS26 directly recruits the

import machinery through its Zn-binding domain.[9] ESS2, which is disordered in the mature 40S, may act as an initial point of attack for Tsr2 to disassemble the importin:eS26 complex. The molecular interactions between Tsr2 and ESS2 described here could allow Tsr2 to target an importin:eS26 complex. Once bound further interactions with ESS1 could facilitate dissociation of the r-protein from the importin. Despite harboring distinct binding sites, eS26 does not simultaneously engage with the importin and Tsr2. Addition of excess importins does not dissociate the Tsr2:eS26 complex (Supplementary Fig. 5a). Therefore, extraction of eS26 by Tsr2 from importin:eS26 complexes cannot be explained by thermodynamic coupling mechanism. It could be that after binding to Tsr2, conformational changes within the Zn-binding domain ensure that the importin binding

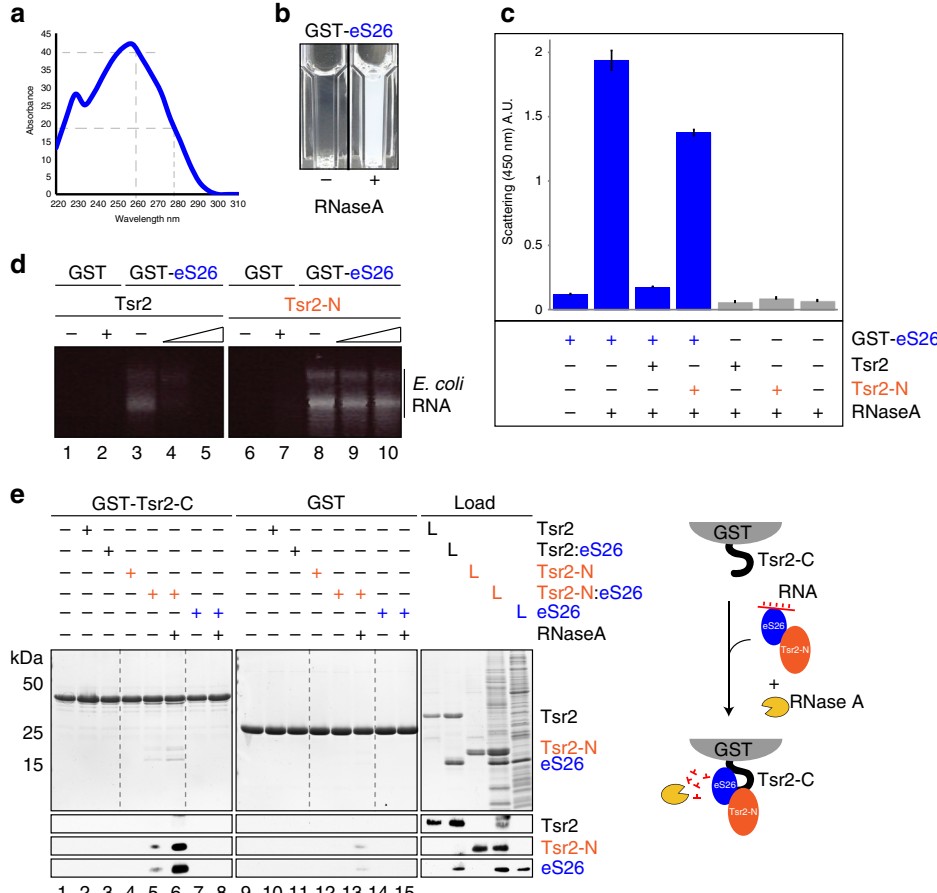

**Fig. 6** C-terminal acidic tail of Tsr2 keeps eS26-RNA free. **a** eS26 co-enriches nucleic acids. **b** RNase A triggers aggregation of eS26. GST-eS26 was treated with RNase A and incubated for 10 min at RT in a photometric cuvette. **c** Tsr2 prevents aggregation of recombinant eS26 in vitro. Thirty-three micromolar GST-eS26 and a two-fold concentration of Tsr2 (66 μM) in PBSKMT was pre-incubated for 1 h at 4 °C (final volume: 90 μl). One microgram of RNase A was added to initiate aggregation. After 1 h of incubation, the scattering signal of the aggregated eS26 was monitored at 450 nm (Y-axes). Three replicates for each well were measured. The error bars show the standard deviation. **d** Tsr2 releases RNA bound to GST-eS26. RNA was extracted from immobilized GST-eS26 after addition of increasing amounts of Tsr2 or Tsr2-N, respectively, separated on a 1% agarose gel and stained by EtBr. **e** GST-Tsr2-C was immobilized on Glutathione Sepharose before incubation with purified Tsr2, Tsr2-N or/and an *E. coli* lysate containing recombinant eS26 in the presence or absence of RNaseA. L = input (1:10 diluted)

site is occluded. Such an allosteric mechanism might guarantee irreversibility to the process of eS26 nuclear import. A comparison of the ribosome-bound eS26 with the NMR data of eS26 in complex with Tsr2 revealed similar secondary structure boundaries for the assigned residues (Supplementary Fig. 3b). This suggests that eS26 when bound to Tsr2 adopts a native-like conformation that resembles the 40S-bound state, thus poising the r-protein for incorporation into the pre-ribosome (Supplementary Fig. 3c).

Archaeal eS26 does not interact with both importins and Tsr2 in vitro (Supplementary Fig. 6a and b). Thus, eukaryotic eS26 has acquired features that enable it to interact with the import machinery and with Tsr2. Archaeal eS26 when fused to ESSs can bind to Tsr2. However, this fusion construct is unable to complement the lethality of eS26-depletion (Supplementary Fig. 6c). Attaching a SV-40 NLS to this fusion protein did not complement the lethality of the eS26-depletion strain suggesting that eukaryotic eS26 has acquired additional elements over its archaeal counterpart that are critical for its role during eukaryotic translation.[32] While eukaryotic eS26 orthologues have ESSs, only ESS1, which forms part of the β-sheet structure of eS26, is highly conserved (Supplementary Fig. 6d). In contrast, ESS2 shows variability both in length and in sequence, which might reflect yet

unknown translation specific functions in different organisms. Despite these differences, hTsr2 is able to substantially rescue severe phenotypes exhibited by yeast depleted for Tsr2, suggesting that the hTsr2 is sufficiently promiscuous to interact with yeast ESS2, and perform its role as an escortin.

Dedicated chaperones systems exhibit different protein folds ranging from compact beta-propellers (Sqt1 and Rrb1) to elongated helical ARM-repeat (Syo1), Ankyrin-repeat (Yar1) and TPR-domain (Acl4) folds that present a large surface area to bind to r-proteins.[8,15,19,29] The helix-rich fold of Tsr2 seems to have evolved from the N-terminal domain of the mRNA transport receptor Nab2. Although structural homology was identified between Tsr2-N and Nab2, closer inspection revealed differences; helix α2 of Tsr2 possesses no structural equivalent in Nab2, helix α3 of Tsr2 is N-terminally extended by more than one helical turn and helix α4 of Tsr2-N is C-terminally extended by one helical turn (Supplementary Fig. 2d). Interestingly, the extensions of helices α3 and α4 in Tsr2-N make critical contacts with ESS2, while helix α2 stabilizes these extensions (Fig. 3c–e). Strong chemical shift perturbations upon eS26 binding are seen in these regions (Fig. 2c–e), all of which are absent in Nab2. Thus, Tsr2 may have evolved from the Nab2 fold by acquiring elements that specifically recognize eS26.

Mutations in the human eS26 gene that result in reduced eS26 protein levels have been linked to DBA.[25,33] However, no pathogenic mutation has yet been mapped within ESSs. The eS26C77W DBA mutant interacts poorly with its import receptors, suggesting that the inability to interact with importins may induce degradation of the mutant protein.[9] The eS26D33N DBA mutant interacted efficiently with importins and Tsr2 in vitro.[9] The molecular basis underlying its instability and failure to be incorporated into the pre-ribosome remains unclear. Supporting the idea that mechanisms that maintain correct eS26 levels are critical during human hematopoiesis, a pathogenic X-linked mutation (E64G) was found in human Tsr2. Here, we show that this mutant protein is impaired in binding human eS26, specifically through ESS2. Thus, our study provides a mechanism that underlies the pathogenicity of this DBA mutation. Although the mutated E64 equivalent residue in hTsr2 seems to be a leucine in yeast (L84), it is a negatively charged residue in all the other species including *S. pombe* (Supplementary Fig. 5). L84 does interact with ESS2 in the yeast Tsr2-ESS2 complex, with residues N-terminal to R100. Considering the importance of the electrostatics for Tsr2-ESS2 interactions in this part of the complex (Fig. 3b), it is not surprising that mutating this negatively charged sidechain in hTsr2 impairs interactions with human ESS2.

We found that the C-terminal tail of Tsr2 keeps recombinant eS26 free of RNA in vitro. Interestingly, our NMR data show that in the Tsr2:eS26 complex the acidic tail of Tsr2 still remains rather flexible (Fig. 2c). Possibly, the C-terminal tail dynamically interacts with regions of eS26 destined to make contacts with ribosomal RNA. A similar role has been proposed for the RNA chaperone Hfq, a hexameric Sm protein that facilitates base pairing between bacterial sRNAs and mRNAs involved in stress response and pathogenesis.[34] Like Tsr2, Hfq possesses an acidic intrinsically disordered C-terminal domain that auto-regulates RNA binding to the Sm ring. This mechanism protects against non-specific RNA binding and aggregation, and facilitates RNA release. It is tempting to speculate that the acidic tail of Tsr2 may prevent eS26-RNA-binding sites from undesired premature encounters, while still enabling an efficient release. Structural analyses of the Fap7:uS11 complex, another dedicated chaperone/ r-protein pair present in all archaeal species, showed that Fap7 stabilizes uS11, and functions as an RNA mimic possibly to prepare the RNA binding surface of the r-protein for incorporation into the 90S pre-ribosome.[35] Protecting the RNA-binding site through RNA mimicry could represent a general mechanism to ensure r-proteins do not engage in interactions with other RNAs during their targeting to their functional binding site. In addition to its role in protecting the rRNA-binding site, Fap7 prefabricates a uS11:eS26 complex for incorporation of these r-proteins to the pre-ribosome.[36] This pathway does not depend on ESS2 since, analogous to the Tsr2-depletion mutant strain,[36] growth impairment of the eS26$^{\Delta ESS2}$ expressing

strain can be substantially rescued upon overexpression of both Fap7 and uS11 (Supplementary Fig. 7a). Moreover, the eS26$^{\Delta ESS2}$ mutant protein is efficiently recruited to the Fap7:uS11 complex, and contrary to WT eS26, the mutant eS26$^{\Delta ESS2}$ protein cannot be extracted by Tsr2 from the Fap7:uS11:eS26$^{\Delta ESS2}$ complex (Supplementary Fig. 7b). Given that Fap7, but not Tsr2, is found in anucleate archaea, we speculate that the Fap7:uS11:eS26 complex could be an ancient mechanism to co-ordinate delivery of these r-proteins to the assembling pre-ribosome.

Compartmentalization is a striking feature of eukaryotic cellular organization. Since protein synthesis occurs in the cytoplasm, eukaryotes have evolved transport factors that bind cargos through signal sequences and target them to their correct compartment. However, the fate of the cargos after being released in their functional compartment remains poorly explored. eS26, like many import cargos, is targeted to the nucleus by the nuclear import machinery. After reaching the nucleus, Tsr2 recognizes the importin:eS26 complex through interactions with ESSs, extracts eS26, and channels the r-protein for ribosome assembly (Fig. 7). Thus, in addition to compartment-targeting signals, cargos may contain features that are recognized by local targeting devices, such as escortins, that ensure their safe and timely arrival at their final functional site.

## Methods

**Yeast strains and plasmids**. The *Saccharomyces cerevisiae* strains used in this study are listed in Supplementary Table 4. Genomic disruptions, C-terminal tagging, and promoter switches at genomic loci were performed according to established protocols.[37–39]

Plasmids used in this study are listed in Supplementary Table 5. Details of plasmid construction will be provided upon request. All recombinant DNA techniques were performed according to established procedures using *E. coli* XL1 blue cells for cloning and plasmid propagation. Point mutations in *RPS26A* and *TSR2* were generated using the QuikChange site-directed mutagenesis kit (Agilent Technologies, Santa Clara, CA, USA). All cloned DNA fragments and mutagenized plasmids were verified by sequencing.

**Fluorescence in situ hybridization and microscopy**. Localization of 20S pre-rRNA was analyzed using a Cy3-labeled oligonucleotide probe (5′-Cy3-ATG CTC TTG CCA AAA CAA AAA AAT CCA TTT TCA AAA TTA TTA AAT TTC TT-3′) that is complementary to the 5′ portion of ITS1.[40]

Pre-40S subunit export, monitored by localization of uS5-GFP and localization of GFP-eS26.[40,41]

Cells were visualized using a DM6000B microscope (Leica, Germany) equipped with a HCX PL Fluotar ×63/1.25 NA oil immersion objective (Leica, Solms Germany). Images were acquired with a fitted digital camera (ORCA-ER; Hamamatsu Photonics, Hamamatsu, SZK, Japan) and Openlab software (Perkin-Elmer, Waltham, MA, USA).

**Tandem affinity purifications and western analyses**. Whole-cell extracts were prepared by alkaline lysis of yeast cells.[42] Tandem affinity purifications (TAP) of pre-ribosomal particles were carried out as previously described.[40,41] Calmodulin-eluates were separated on NuPAGE 4–12% Bis-Tris gradient gels (Invitrogen, Carlsbad, CA, USA) and visualized by either Silver staining or western analyses using indicated antibodies.

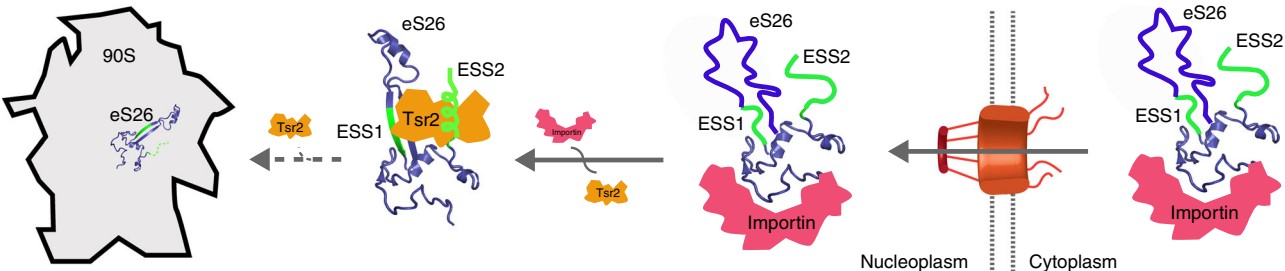

**Fig. 7** Model for the transport of eS26 to the 90S pre-ribosome. Newly synthesized and unfolded eS26 is transported from the cytoplasm into the nucleus by importins. In the nucleus, Tsr2 dissociates eS26 from importins in a RanGTP-independent manner through eukaryotic-specific segments (ESSs) in eS26. Tsr2 then sequesters the released eS26 enabling safe transfer to the 90S pre-ribosome

Western analyses were performed as previously described.[42] The following primary antibodies were used in this study: α-Tsr2/S26 (1:3000; this study), α-Arc1 (1:4000; E. Hurt, University of Heidelberg, Heidelberg, Germany), α-uL3 (yeast Rpl3) (1:5000; J. Warner, Albert Einstein College of Medicine, Bronx, NY, USA), α-uS7 (yeast Rps5) (1:4000; Proteintech Group Inc., Chicago, IL, USA, custom-made), α-uS3 (yeast Rps3) (1:3000; M. Seedorf, University of Heidelberg, Heidelberg, Germany); α-TAP (CBP) (1:4000; Thermo Scientific, Rockford, IL, USA, cat. no.: PA5-27369), α-Pno1 (1:10,000; K. Karbstein, Scripps Research Institute, Jupiter, FL, USA), α-Dim1 (1:10,000; K. Karbstein, Scripps Research Institute, Jupiter, FL, USA), α-Nob1 (1:500; Proteintech Group Inc., Chicago, IL, USA, custom-made), α-Tsr1 (1:10,000; K. Karbstein, Scripps Research Institute, Jupiter, FL, USA), α-Ltv1 (1:5000; K. Karbstein, Scripps Research Institute, Jupiter, FL, USA), α-Rio2 (1:1000; Proteintech Group Inc., Chicago, IL, USA, custom-made), α-FLAG (1:3000; Sigma-Aldrich, St. Louis, MO, USA, F3165). The secondary HRP-conjugated α-rabbit and α-mouse antibodies (Sigma-Aldrich, USA, cat. no: A0545, A4416) were used at 1:1000–1:5000 dilutions. Protein signals were visualized using the Immun-Star HRP chemiluminescence kit (Bio-Rad Laboratories, Hercules, CA, USA) and captured by Fuji Super RX X-ray films (Fujifilm, Tokyo, Japan). Uncropped blots are shown in Supplementary Fig. 8.

**Recombinant protein expression and binding assays.** All recombinant proteins were expressed in E. coli BL21 cells by IPTG induction. His6-tagged proteins were affinity purified in 50 mM Hepes pH 7.5, 50 mM NaCl, 10% glycerol using Ni-NTA Agarose (GE Healthcare), GST-fusion proteins were purified in PBSKMT (150 mM NaCl, 25 mM sodium phosphate, 3 mM KCl, 1 mM MgCl2, 0.1% Tween, pH 7.3) using Glutathione Sepharose (GE Healthcare).

Recombinant GST-Tsr2 was immobilized in PBSKMT (high—500 mM NaCl addition and low—no NaCl addition salt conditions) on Glutathione Sepharose (GE Healthcare), and incubated with E. coli lysates containing recombinant eS26, eS26FLAG, eS26ΔESS1-FLAG, eS26ΔESS2-FLAG, and eS26ΔESS1ΔESS2-FLAG for 1 h at 4 °C. After incubation, the immobilized GST-proteins were washed three times with 1 ml of PBSKMT 4 °C. The bound proteins were eluted with LDS. The in vitro binding studies between recombinant eS26FLAG, eS26ΔESS1-FLAG, eS26ΔESS2-FLAG, eS26ΔESS1ΔESS2-FLAG, Tsr2, Tsr2:eS26 complex, and yeast importins as GST-fusion proteins were performed as previously described[43]. One-fifth of the bound proteins and input (eS26, eS26FLAG, eS26ΔESS1-FLAG, eS26ΔESS2-FLAG, and eS26ΔESS1ΔESS2-FLAG) were analyzed on a Coomassie Blue-stained gel. One-tenth of the bound proteins and 1/1000th of the input was used for western analyses.

To dissociate the GST-importin (Kap123):eS26FLAG, eS26ΔESS1-FLAG, eS26ΔESS2-FLAG, and eS26ΔESS1ΔESS2-FLAG complex in RanGTP-independent manner, 1.5 μM His6-Tsr2 was added to preassembled complexes for 1 h at 4 °C protocol modified from refs. [44,45,46]. Bound proteins were eluted in two-fold LDS/SDS sample buffer by incubating at 70–95 °C and separated by SDS-PAGE. Proteins were visualized by Coomassie Blue staining or by western analyses using antibodies against FLAG tag of the fusion protein variants of eS26.

**RNA extraction and analysis.** RNA was extracted from immobilized GST-eS26 or flow-through sample after incubation with increasing amounts of Tsr2. RNA was extracted with phenol–chlorofom–isoamylalcohol and precipitated in isopropanol. RNA pellets were washed with 80% ethanol and resuspended in 20 μl water. rRNAs were then separated on a 1.2% Agarose/formaldehyde gel for 1.5 h at 200 V.

**XL-MS.** Crosslinking with DSS-d0/d12 (Creative Molecules), ADH-do/d10 and PDH-do/d10 and DMTMM (all Sigma-Aldrich) was performed following previously published procedures.[21,22] Briefly, the protein complex was prepared at a concentration of 1 mg/ml in 50 mM HEPES. DSS crosslinking was performed at two different concentrations of the reagent (0.5 and 1 mM) for 30 min at 37 °C. PDH/DMTMM crosslinking was performed at final concentrations of 9 and 12 mg/ml of PDH and DMTMM, respectively, for 45 min at 37 °C. One hundred micrograms of total protein was used for each experiment, the crosslinked samples were split in half, and 50 μg were digested with endoprotease Lys-C (Wako) and trypsin (Promega) or, alternatively, with endoprotease Glu-C (Roche). An additional experiment was performed using ADH instead of PDH, and using only Lys-C/trypsin for digestion.

Digests of crosslinked proteins were fractionated by size-exclusion chromatography (Superdex Peptide PC 3.2/300; GE Life Sciences) as described previously,[47] and fractions were analyzed by liquid chromatography/tandem mass spectrometry on a Thermo Orbitrap Elite or Orbitrap XL instruments.[48] Data analysis was performed using xQuest[49] by searching against a sequence database of the target proteins and contaminants. Only crosslinks identified on target proteins were considered further. The selected score threshold corresponds to a false discovery rate of less than 5% for both intra- and inter-protein crosslinks.

**Protein expression and purification for NMR analyses.** All protein constructs were expressed as C-terminal fusions to a TEV protease-cleavable (His)6-tagged GB1 domain in E. coli BL21 (DE3) CodonPlus-RIL cells (Stratagene). The employed expression vector pEM1 was generated by subcloning the XbaI/BamHI fragment of the cell-free expression vector pCFX3 (ref. [50]) into pET19b (Novagen, Madison, WI, USA). Unlabeled or uniformly 15N- and 13C,15N-labeled proteins

were obtained by growing the cells in 2 l of LB or M9 medium, respectively.[51] The cell cultures were grown at 37 °C to an OD600 of ca. 0.5, then transferred to 20 °C before inducing expression with 1 mM IPTG at an OD600 of ca. 0.8. The cells were harvested after 16 h by centrifugation (15 min at 4000 g and 4 °C) and were resuspended in 30 ml buffer A (50 mM Tris–HCl at pH 7.6, 500 mM NaCl, 20 mM imidazole, 1 mM DTT) on ice. Cells were disrupted with five passages at 75 psi through an ice-cooled M-110 L cell cracker (Microfluidics, Westwood, MA, USA) and the lysate was cleared by centrifugation (30 min at 30,000 g and 4 °C). The supernatant was applied on a 5 ml HisTrap HP column (GE Healthcare), washed with 10 column volumes of buffer A, and target proteins were eluted in a 100 ml linear gradient of 20–500 mM imidazole in buffer A. The obtained fractions were analyzed by SDS-PAGE and target protein-containing fractions were pooled, supplied with 2 mg (His)6-tagged TEV protease,[1,50] and dialyzed for 20 h at 4 °C in a 3.5 kDa molecular weight cutoff (MWCO) Spectra/Por 3 dialysis membrane (Spectrum Labs, CA, USA) against 4 L cleavage buffer (50 mM Tris–HCl, pH 7.6, 100 mM NaCl, 1 mM DTT). With the exception of the ESS2 peptides, the cleaved (His)6-GB1 domain and the (His)6-tagged TEV protease were separated from the target proteins by another passage over a 5 ml HisTrap HP column in buffer A. The fractions of the flow-through containing the target protein were detected by SDS-PAGE, pooled, and concentrated in a 3 kDa MWCO Amicon Ultra-15 centrifugal filter device. The concentrated sample was purified on a Superdex 75 HiLoad 16/60 column (GE Healthcare) equilibrated in NMR buffer (20 mM sodium phosphate, pH 7.0, 1 mM DTT, 10 μM EDTA) and the fractions containing the target protein were concentrated to 0.5–0.8 mM and supplemented with 5% (v/v) D2O before NMR measurements. The ESS2 peptides were separated from the (His)6-GB1 domain and the (His)6-tagged TEV protease by reverse-phase HPLC using a preparative C8 column (Agilent). The TEV cleavage reaction was supplemented with 0.1% TFA and 2.5% acetonitrile and was applied on the C8 column equilibrated in buffer TA (water containing 0.1% (v/v) TFA and 2.5% (v/v) acetonitrile). The ESS2 peptide was then separated from the proteins using a linear gradient from 2.5–100% acetonitrile in buffer TA. The fractions containing ESS2 were validated by LC-MS and were lyophilized to remove acetonitrile and TFA. The dried peptide was dissolved in NMR buffer containing 5% (v/v) D2O and was titrated to pH 7.0 with NaOH. NMR samples of Tsr2-N in complex with ESS2 were prepared by stepwise titration of the unlabeled component to the labeled one while following the complex formation with 2D 15N,1H-HSQC spectra until no amide signals of the free state were detected. Two samples with isotope labeling of either Tsr2-N or ESS2 with the other component unlabeled were prepared for NMR analysis in this way.

**NMR analysis and structure determination.** All NMR experiments were recorded on Bruker Avance 500, 600, 700, and 900 MHz spectrometers equipped with a triple-resonance CryoProbeTM with shielded z-gradient coils. Samples of Tsr2-N and Tsr2-N in complex with ESS2 were measured at 293.15 K while those of Tsr2 and Tsr2 in complex with eS26 were measured at 303.15 K. Quadrature-detection in the indirect dimensions was achieved by States time-proportional phase incrementation.[52] The water signal was suppressed with spin-lock pulses or WATERGATE.[53] The raw NMR data were processed with TOPSPIN 3.0 (Bruker, Billerica, MA). Proton chemical shifts are referenced to the water resonance and 13C and 15N chemical shifts are indirectly referenced to 1H using the absolute frequency ratios.[54] Backbone resonances were assigned with 3D HNCA,[55] 3D HNCACB,[55] 3D CBCA(CO)NH[56] experiments while sidechain resonances were assigned using the 3D (H)CC(CO)NH,[57] 3D H(CC(CO)NH[57] and 3D [15N,1H]-HSQC-TOCSY[58] experiments using samples containing 0.5–0.8 mM of uniformly 13C,15N-labeled protein in NMR buffer. Heteronuclear 15N{1H} nuclear Overhauser effect (NOE) experiments[59] were recorded at 700 MHz with a relaxation delay of 5 s and a 3 s saturation period in the saturation experiment. Chemical shift perturbations were calculated from combined amide proton and nitrogen chemical shift differences, $((\Delta\delta^1H)^2 + (0.2 \cdot \Delta\delta^{15}N)^2)^{1/2}$, where $\Delta\delta^1H$ and $\Delta\delta^{15}N$ indicate the residue-specific chemical shift differences of the amide proton and nitrogen in the free and bound states, respectively. Secondary structure boundaries of proteins were analyzed using the assigned 13Cα shifts according to the chemicals shift index protocol.[60] NOE-based distance constraints for the structure calculation were obtained from 3D 15N-resolved [1H,1H]-NOESY, 3D aliphatic 13C-resolved [1H,1H]-NOESY, and 3D aromatic 13C-resolved [1H,1H]-NOESY spectra,[61,62] which were recorded with a mixing time of 60 ms. Two sets of NOE-based distance constraints were obtained from the NOESY spectra measured with two samples of Tsr2-N in complex with ESS2 with either Tsr2-N or ESS2 13C,15N-labeled. The protocol for the NMR structure calculation employed the ATNOS/CANDID[63] procedure for automated peak picking in combination with the chemical shift lists from the sequence-specific resonance assignments for Tsr2-N and ESS2 and the sets of 3D NOESY spectra as input for automated NOESY assignment and structure calculation in the program CYANA.[64] The final CYANA NOE assignment and structure calculation of Tsr2-N in complex with ESS2 employing the CYANA macro NOEASSIGN, and used the lists of picked peaks from cycle 2 of the ATNOS/CANDID procedure together with a manually defined list of 28 upper distance limits based on a set of weak peaks in the NOESY spectra which were not assigned by ATNOS/CANDID or NOEASSIGN. The final structure calculation in cycle 7 included only unambiguously assigned distance constraints based on the calculated 3D structure from cycle 6. The 20 conformers with the lowest residual

target function obtained from cycle 7 were then energy-minimized in implicit water using the program AMBER12 (ref. [65]) For spectral analysis the program CARA (www.nmr.ch) was used.[66] The programs MOLMOL,[67] PyMOL (Schrödinger, LLC), and the UCSF Chimera package[68] were used for visualization of the protein structures. Molecular surfaces were calculated using MSMS.[69]

**Isothermal titration calorimetry**. ITC experiments for determining the affinity between Tsr2 and ESS2 were conducted on a VP-ITC instrument (MicroCal Inc., Northampton, MA). The experimental setup comprised 1.4 ml of 20–40 µM Tsr2 variant in the sample cell and 400–800 µM of the ESS2 peptide in the syringe. Measurements were performed at 25 °C in NMR buffer (20 mM NaPi, pH 7.1, 1 mM DTT, 10 µM EDTA) with a stirring rate of 300 rpm, and comprised 29 injections with 10 µl volume of 10 s duration with 4 min intervals between injections. The raw data were integrated, corrected for non-specific heats, normalized for concentration, and analyzed using the MicroCal Origin software.

**Circular dichroism (CD) spectroscopy**. Far-UV CD spectra were recorded on a JASCO J-810 spectropolarimeter at 20 °C, using 50 µM of the Tsr2 variants in NMR buffer in 1 mm QS quartz cuvettes (Hellma Analytics) and scanning from 260 to 190 nm. The results were displayed as molar ellipticity $[\theta]$ with units of [deg $cm^2$ $dmol^{-1}$]. The thermal denaturation curves were obtained by following the molar ellipticity at 222 nm while increasing the temperature by 1 °C per minute from 20 to 93 °C.

## Data availability

Structural data have been deposited in the PDB with the following accession codes: Tsr2 (1-152): 6G03, Tsr2(1-152)-S26A(100-119): 6G04. Crosslinking mass spectrometry data have been deposited to the ProteomXchange Consortium via the PRIDE partner repository with the dataset identifier PXD009106.

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

## Acknowledgements

V.G.P. is supported by grants from the Swiss National Science Foundation, NCCR RNA & Disease, Novartis Foundation, Olga Mayenfisch Stiftung and a Starting Grant Award from the European Research Council (EURIBIO260676). M.O. is supported by a PhD Fellowship from the Boehringer Ingelheim Fonds. F.H.-T.A. acknowledges support from the Swiss National Science Foundation and NCCR RNA & Disease. R.A. is supported by ERC grant Proteomics v3.0 (AdvG233226).

## Author contributions

S.S., C.P., and V.G.P. designed the study. S.S., M.O., and C.P. performed yeast functional studies. S.S., E.M., and F.F.D. performed NMR studies. A.L. performed XL-MS studies in the laboratory of R.A. S.S., E.M., F.F.D., C.P., F.H.-T.A., and V.G.P. wrote the manuscript with input from remaining authors.

## Additional information

**Competing interests:** The authors declare no competing interests.

