## [Peer Review File · Nature Communications]

Reviewers' comments:

Reviewer #1 - Expertise: Ribosome biogenesis, nuclear import (Remarks to the Author):

In their 2014 paper, the Panse group reported that Tsr2 is a chaperone of eS26, which acts after nuclear import of eS26 and disrupts the complex between the ribosomal protein and importins. In 2016, they followed up on that story and reported a role of Fap7/eS11 in eS26 assembly, which might either represent an alternative eS26 assembly pathway or a later step in the Tsr2-dependent assembly route.

In the present study, they further extend the characterization of Tsr2 by solving its NMR structure and mapping the Tsr2 binding site within eS26. Interestingly, similar to other described ribosomal protein chaperones, eukaryote specific sequences (ESS) in eS26 are critical for the interaction with Tsr2. The authors further show that disruption of the complex by either removing the eS26 ESSs, or by mutating residues in Tsr2 results in phenotypes similar to the ones they previously observed upon depletion of either protein.

Moreover, they show that eS26 lacking the ESSs is only inefficiently extracted by Tsr2, confirming that the interaction via these sequences is necessary for the function of Tsr2 in releasing eS26 from importin.

The quality of the experiments is generally high and the manuscript is easy to read and follow. However, especially the introduction and the discussion are in parts very superficial and relevant studies from other groups, but even from their own group are not even mentioned.

Furthermore, some of the data should be supported by additional experiments.

Suggestions:

- if the model the authors propose is correct and eS26 Δ ESS2 is imported into the nucleus but not dissociated from importin, the protein should become trapped in the nucleus in complex with importin. Can the authors show that this is the case?

- a high number of nuclear transport papers are cited (which is fine), while almost no papers dealing with ribosome biogenesis, or with dedicated ribosomal protein chaperones are cited (and the single paper that is cited doesn't even contain all of the data mentioned in the sentence before).

To be able to understand the "big picture", comparison of the data to other ribosomal protein/chaperone complexes would be useful (where it makes sense), e.g. with respect to binding to ESS, structural features (e.g. alpha-helical structures) involved in interaction with the ribosomal protein, etc

Even more dubious to me is the fact that the authors do not even mention, nor cite their 2016 paper on Tsr2/eS26. In that paper they showed some functional redundancy between Tsr2 and Fap7, a chaperone for uS11 and reported about prefabrication of a eS26/uS11 complex (together with Fap7). This is critical information that has to be discussed (also in context with the new results) in order to give the reader the chance to understand how eS26 is delivered to 40S precursors. There are only two papers dealing with eS26 delivery/incorporation (both from the same group) and the omission of one of them leaves the reader with the feeling that the authors want to hide something.

- "Yet, how Tsr2 disassembles only importin:eS26 complexes amongst diverse importin:cargo complexes entering the nucleus remains unresolved"

Really? The authors showed in their 2014 paper that eS26 is the only interaction partner of Tsr2, therefore it is obvious that it won't disassemble any other importin: cargo complexes

- the study unravels the importance of eS26 ESS2. Does it just function as a recruitment sequence for Tsr2, or does it also have other functions, e.g. in the ribosome, or in binding the Fap7/uS11 complex? Overexpression of Fap7/uS11 in the delta ESS2 strain may help to better understand this aspect. In their 2016 paper, the authors showed that this can rescue the *tsr2* deletion phenotypes, so it would be important to know if Fap7/uS11 can also replace the function of Tsr2 when the eS26 ESS2 is missing.

- what is the conclusion from the structures? The structure is mainly used to choose the residues in Tsr2 to be mutated for further phenotypic analysis, but otherwise, no conclusion with respect to the mechanism of action of Tsr2 is drawn. Even though the authors were not able to solve the full length Tsr2/eS26 complex structure, I'm sure that some interpretation would be possible. A model of full length eS26 in complex with Tsr2 (with the help of XL-MS data) would be helpful, especially in comparison with the structure of eS26 bound to the ribosome and with a modeled eS26/eS11/Fap7 complex. These analyses may also help support the hypothesis raised in the discussion that Tsr2 could act as an rRNA mimic.

- the Tsr2 DBA mutant protein shows reduced binding to eS26. The authors conclude that in the disease, the importin extraction mechanism is inhibited. But how can the suppression by overexpression of eS26 then be explained? Shouldn't this result in even more eS26 becoming trapped with importin? Does eS26 overexpression also rescue *tsr2* depletion or DWI mutation? In the 2014 study, the authors already analyzed two DBA related mutants in eS26, which they do not mention here. Do the mutations target the same step in the assembly pathway? This should be discussed.

Minor points:

- one can somehow deduce it indirectly from the Figures and the name of the protein, but the authors should also explicitly mention that the protein does not exist in bacteria (and if it is present in some or all archaea).

Reviewer #2- Expertise: Solution NMR, Protein-Protein Interactions (Remarks to the Author):

This is a solid paper describing the structure of the escortin Tsr2 alone and in complex with an intrinsically disordered peptide derived from the ribosomal protein Es26. The NMR work looks solid, and the ancillary biochemistry seems fine. The results are interesting and of general interest and I recommend acceptance.

Minor comment.

The right panel in Fig. 2b and Fig. 3b would look a lot clearer if protons were omitted from the structures.

Reviewer #1 - Expertise: Ribosome biogenesis, nuclear import
(Remarks to the Author):

In their 2014 paper, the Panse group reported that Tsr2 is a chaperone of eS26, which acts after nuclear import of eS26 and disrupts the complex between the ribosomal protein and importins. In 2016, they followed up on that story and reported a role of Fap7/eS11 in eS26 assembly, which might either represent an alternative eS26 assembly pathway or a later step in the Tsr2-dependent assembly route.

In the present study, they further extend the characterization of Tsr2 by solving its NMR structure and mapping the Tsr2 binding site within eS26. Interestingly, similar to other described ribosomal protein chaperones, eukaryote specific sequences (ESS) in eS26 are critical for the interaction with Tsr2. The authors further show that disruption of the complex by either removing the eS26 ESSs, or by mutating residues in Tsr2 results in phenotypes similar to the ones they previously observed upon depletion of either protein.

Moreover, they show that eS26 lacking the ESSs is only inefficiently extracted by Tsr2, confirming that the interaction via these sequences is necessary for the function of Tsr2 in releasing eS26 from importin.

The quality of the experiments is generally high and the manuscript is easy to read and follow. However, especially the introduction and the discussion are in parts very superficial and relevant studies from other groups, but even from their own group are not even mentioned.

Furthermore, some of the data should be supported by additional experiments.

Suggestions:

- if the model the authors propose is correct and eS26 Δ ESS2 is imported into the nucleus but not dissociated from importin, the protein should become trapped in the nucleus in complex with importin. Can the authors show that this is the case?

We have attempted to address the suggestion by co-IP analyses. For this, we immuno-precipitated ProteinA-FLAG-eS26 and ProteinA-FLAG-eS26 Δ ESS2 and monitored importin (Pse1 and Kap123) co-enrichment by Western blotting. We were unable to co-enrich detectable levels of Pse1 and Kap123 in these IPs. A possible explanation could be that at steady state only a small amount of eS26, below the detection levels of our Western analyses, is bound to its importin during transfer to the nucleus. Alternatively, it is possible that the importin:eS26 complexes are not stable, and fall apart upon cell lysis.

Rebuttal Figure 1: Whole cell extracts (WCE) and FLAG-tag co-immunoprecipitations were prepared from cells expressing FLAG-tagged eS26 or eS26 Δ ESS2 and subjected to Western analysis using antibodies directed against FLAG-tag, Kap123 and Pse1

We would like to clarify our model regarding Tsr2 function that was presented in Schütz et al., 2014 *eLife*. In that study, we showed that the importin:eS26 complexes, unlike a typical importin:cargo complex, was inefficiently dissociated by RanGTP *in vitro*. Instead, Tsr2, without RanGTP assistance, efficiently releases eS26 from its importin, and prevents its aggregation *in vitro*. Tsr2-depletion *in vivo* renders eS26 susceptible to proteolysis, precluding its incorporation into the 90S pre-ribosome. Based on a combination of these *in vitro* and *in vivo* findings we proposed that Tsr2 extracts eS26 from its importin in a RanGTP-independent manner, and ensures safe transfer of the cargo to the 90S. Failure to extract eS26 from its importin *in vivo* seems to trigger degradation of the bound ribosomal protein cargo via a yet unknown mechanism/factor. We speculate that competing Tsr2 unloading activity and degradation activity ensures that importins are freed from the cargo and recycled back to the cytoplasm for the next round of nuclear import.

- a high number of nuclear transport papers are cited (which is fine), while almost no papers dealing with ribosome biogenesis, or with dedicated ribosomal protein chaperones are cited (and the single paper that is cited doesn't even contain all of the data mentioned in the sentence before).

We have now cited reviews on ribosome assembly and primary literature related to dedicated chaperone systems.

To be able to understand the "big picture", comparison of the data to other ribosomal protein/chaperone complexes would be useful (where it makes sense), e.g. with respect to binding to ESS, structural features (e.g. alpha-helical structures) involved in interaction with the ribosomal protein, etc.

We have compared our findings to other dedicated chaperone ribosomal

protein complexes (Please see Discussion).

Recently, dedicated chaperones Syo1, Sqt1, Rrb1, Yar1 and Acl4 were proposed to capture their respective ribosomal protein clients in the cytoplasm during translation *via* their eukaryotic specific N-terminal segments. uL4 is an exception to this wherein a eukaryotic specific loop insertion appears to provide the binding platform for Acl4 for co-translational capture. In addition to co-translational capture Syo1, Rrb1, Yar1 and Acl4 function as adaptors to recruit the nuclear import machinery. In contrast, to these ribosomal protein/dedicated chaperone systems, eS26 directly recruits the import machinery through its Zn-binding domain. The C-terminal eukaryotic specific extension, ESS2, functions to attract Tsr2 to the importin:eS26 complex to trigger RanGTP independent disassembly.

Structural studies of dedicated chaperones systems revealed that they exhibit different protein folds ranging from compact beta-propellers (Sqt1 and Rrb1) to elongated helical ARM-repeat (Syo1), Ankyrin-repeat (Yar1) and TPR-domain (Acl4) that present a large surface area to bind to ribosomal proteins and prevent them from engaging in non-productive interactions. Curiously, the helix-rich fold of Tsr2 seems to have evolved from the N-terminal domain of the mRNA transport receptor Nab2.

Even more dubious to me is the fact that the authors do not even mention, nor cite their 2016 paper on Tsr2/eS26. In that paper they showed some functional redundancy between Tsr2 and Fap7, a chaperone for uS11 and reported about prefabrication of a eS26/uS11 complex (together with Fap7). This is critical information that has to be discussed (also in context with the new results) in order to give the reader the chance to understand how eS26 is delivered to 40S precursors. There are only two papers dealing with eS26 delivery/incorporation (both from the same group) and the omission of one of them leaves the reader with the feeling that the authors want to hide something.

RPS26 knockout is lethal in budding yeast, whereas Tsr2-depleted cells are viable but severely growth impaired. This implies at least two mechanisms to unload eS26 from the import machinery and ensure viability of the Tsr2-depletion mutant strain (Schütz et al., 2014 eLife; Pena et al., 2016 eLife).

The major mechanism to extract eS26 from the importin:eS26 complex involves Tsr2. Tsr2 disassembles an importin:eS26 complex and then binds to the released eS26 *in vitro*. Subsequently, eS26 is released from Tsr2, and then recruited to a preformed Fap7:uS11 complex for delivery to the assembling pre-ribosome (Schütz et al., 2014 eLife).

The second Tsr2-independent mechanism employs a pre-formed Fap7:uS11 complex for eS26 extraction from its importin. In this case, the Fap7:uS11 is recruited to eS26 bound to its importin forming a importin:eS26:uS11:Fap7 complex. Interaction of this complex with with RanGTP releases a Fap7:uS11:eS26 complex from the importin (Pena et al. 2016). Both mechanisms converge to form a Fap7:uS11:eS26 complex, a critical step for simultaneous incorporation of both ribosomal proteins (uS11 and eS26) into the pre-ribosome.

The present work deals with how Tsr2 releases eS26 from importin:eS26

complexes. We show that ESS2 plays a critical role to recruit Tsr2 to the importin:eS26 complex to trigger RanGTP independent assembly *in vitro*. Our analyses do not provide insights into how Fap7:uS11 complex recruitment renders an importin:eS26 interactions RanGTP-sensitive.

- “Yet, how Tsr2 disassembles only importin:eS26 complexes amongst diverse importin:cargo complexes entering the nucleus remains unresolved” Really? The authors showed in their 2014 paper that eS26 is the only interaction partner of Tsr2, therefore it is obvious that it won't disassemble any other importin:cargo complexes.

We agree with the Reviewer that the question we addressed in the current manuscript is not precisely phrased. We have corrected this to: “How Tsr2 recognizes an importin:eS26 complex amongst diverse importin:cargo complexes entering the nucleus remains unclear.”

- the study unravels the importance of eS26 ESS2. Does it just function as a recruitment sequence for Tsr2, or does it also have other functions, e.g. in the ribosome, or in binding the Fap7/uS11 complex? Overexpression of Fap7/uS11 in the delta ESS2 strain may help to better understand this aspect. In their 2016 paper, the authors showed that this can rescue the *tsr2* deletion phenotypes, so it would be important to know if Fap7/uS11 can also replace the function of Tsr2 when the eS26 ESS2 is missing.

Previously, we have shown that Fap7:uS11 recruits eS26 through tertiary interactions found between these ribosomal proteins on a mature 40S subunit (Pena et al eLife 2016). As requested, we tested whether ESS2 contributes to eS26 recruitment to the Fap7:uS11 complex (Supplementary Fig. 7). We found that the eS26 Δ ESS2 mutant protein is efficiently recruited to the Fap7:uS11 complex. Further, in contrast to WT eS26, we found that this mutant protein cannot be extracted by Tsr2 from a preformed Fap:uS11:eS26 Δ ESS2 complex. Together, these data support the conclusion that ESS2 does not contribute to the interactions between eS26 and Fap7:uS11 within the Fap7:uS11:eS26 complex. Moreover, like in the case of the Tsr2-depletion mutant strain, growth impairment of the eS26 Δ ESS2 mutant strain can be substantially rescued by overexpression of Fap7 and uS11. These genetic data support the idea that Fap7:uS11 can replace Tsr2-function when the binding platform of Tsr2 within eS26 is missing.

- what is the conclusion from the structures? The structure is mainly used to choose the residues in Tsr2 to be mutated for further phenotypic analysis, but otherwise, no conclusion with respect to the mechanism of action of Tsr2 is drawn. Even though the authors were not able to solve the full length Tsr2/eS26 complex structure, I'm sure that some interpretation would be possible. A model of full length eS26 in complex with Tsr2 (with the help of XL-MS data) would be helpful, especially in comparison with the structure of eS26 bound to the ribosome and with a modeled eS26/eS11/Fap7 complex. These analyses may also help support the hypothesis raised in the discussion that Tsr2 could act as an rRNA mimic.

The major conclusion derived from this study is a detailed description of how ESS2 is recognized by Tsr2. It furnishes a basis for understanding how the eukaryotic specific sequence (not seen in the crystal structure of 40S subunit) allows Tsr2 to target eS26 while bound to the importin. Since the importin does not require the ESSs to bind eS26 this site is available for an initial attack by Tsr2 within the importin:eS26 complex. Once bound further interactions with eS26 could weaken the interaction with the importin facilitating release. We have added text to the discussion to bring out the impact of the structure on understanding of Tsr2 function.

We agree with the reviewer that it would be interesting to know more about the interactions of Tsr2 with eS26 particularly ESS1. We have made an attempt to model the complex of full length eS26 bound to Tsr2 using the coordinates of eS26 (1-98) from the 40S ribosome structure (PDB 4V88), the NMR constraints of the structure of ESS2 bound to Tsr2-N, and distance constraints derived from the crosslinks obtained with mass-spec of the eS26/Tsr2 complex using CYANA. Based upon our NMR data of eS26 bound to Tsr2 and the prediction of disorder for residues at the N- and C-termini of eS26 we allowed residues 1-7 and 91-98 to be flexible. CYANA target functions of the generated models were comparable to calculations performed for Tsr2-N/ESS2 indicating that all assumptions could be fulfilled. All crosslinks to ESS2 were easily satisfied whereas the crosslinks from Tsr2 to eS26 (1-98) were just under the maximum allowed distance in most models. Due to the low distance precision of the mass-spec crosslink constraints (<30Å for DSS LYS-LYS, <21Å for ADH GLU-LYS, <25Å for ZL GLU-LYS crosslinks), a wide range of orientations for eS26 were compatible with the experimental data preventing conclusions from being drawn regarding additional interactions in the full length Tsr2:eS26 complex (See Rebuttal Figure 2).

Rebuttal Figure 2: Models of eS26 bound to Tsr2-N generated using fixed geometry of eS26 residues 8-90 corresponding to coordinates of eS26 in the 40S ribosome (PDB 4V88), constraints from the NMR structure of Tsr2-N:ESS2, and crosslink constraints from XL-MS analysis of the Tsr2:eS26 complex. Tubes correspond to spline fit through C α atoms: Tsr2-N in orange, ESS2 of eS26 in green, and different models representing the four most prevalent orientations of eS26(1-98) with respect to Tsr2-N present in the CYANA ensemble shown in different colors.

Does the Tsr2 C-terminal tail bind to the RNA binding site of eS26? While our biochemical data support this possibility, XL-MS did not reveal any crosslinks between this region and the RNA binding site of eS26, precluding modelling of this interaction within the Tsr2:eS26 complex. Further, NMR analyses revealed that even within the Tsr2:eS26 complex the C-terminal tail is highly dynamic. These data provide explanation as to why we are unable to obtain crosslinks as well as unable to detect intermolecular and intra-molecular NOEs of the C-terminal acidic to define the structure of this region in complex with eS26.

- the Tsr2 DBA mutant protein shows reduced binding to eS26. The authors conclude that in the disease, the importin extraction mechanism is inhibited. But how can the suppression by over-expression of eS26 then be explained? Shouldn't this result in even more eS26 becoming trapped with importin?

Our idea regarding Tsr2 function is based on a combination of *in vitro* and *in vivo* data (Schütz et al., 2014 *eLife*). We showed that importin:eS26 complexes, unlike typical importin:cargo complexes, were inefficiently dissociated by RanGTP *in vitro*. Instead, we found that the escortin Tsr2, without the assistance from RanGTP, efficiently releases eS26 from its importin and prevents its aggregation *in vitro*. Tsr2-depletion *in vivo* renders eS26 susceptible to proteolysis, precluding eS26 incorporation into the 90S pre-ribosome. We proposed that Tsr2 triggers the release of eS26 from its importin in the nucleus and couples the process of eS26 nuclear import and subsequent transfer to the 90S. We infer that failure to extract the eS26 from importin *in vivo* triggers the degradation of the bound cargo ribosomal protein via yet unknown mechanism.

As the Reviewer correctly points out the suppression might operate through law of mass action. Increased levels of importin:eS26 complexes probably compensate for the reduced affinity between Tsr2-DBA mutant and eS26, and hence overcomes the competing eS26 degradation pathway. eS26 that fails to be unloaded by Tsr2 from the importin:eS26 complex would be rapidly degraded.

Does eS26 overexpression also rescue *tsr2* depletion or DWI mutation?

The observed suppression by eS26 is specific to the Tsr2-DBA mutant protein (Figure 5a). Overexpression of eS26 (from a 2 μ high-copy plasmid) does not rescue the growth defect of the Tsr2-depletion and the Tsr2-DWI mutant strain (Rebuttal Figure 3).

Rebuttal Figure 3: The conditional $P_{GAL1-TSR2}$ strain was transformed with the indicated 2 μ high-copy plasmids and spotted in 10-fold dilutions on repressive glucose-containing media and grown at 30°C for 4 days.

In the 2014 study, the authors already analyzed two DBA related mutants in eS26, which they do not mention here. Do the mutations target the same step in the assembly pathway? This should be discussed.

We have now mentioned the eS26 mutations (Please see Discussion section). Two mutations within human eS26, D33N and C77W have been linked to DBA. Both mutations are not within ESS1 and ESS2 regions. The eS26C77W mutant protein interacts poorly with its import receptors, suggesting that the inability to interact with importins may induce degradation of the mutant protein. However, this eS26C77W mutant protein bound to Tsr2 (Schütz et al. eLife 2014). Cysteine 77 is one of four conserved cysteines within eS26 that coordinates a Zn²⁺ ion. The eS26D33N mutant interacted efficiently with both importins and Tsr2 *in vitro* (Figure 6 in Schütz et al., 2014). The molecular basis underlying its instability and failure of this mutant protein to be incorporated into the 90S pre-ribosome remains unclear.

Minor points:

- one can somehow deduce it indirectly from the Figures and the name of the protein, but the authors should also explicitly mention that the protein does not exist in bacteria (and if it is present in some or all archaea).

As requested, we have now explicitly stated that eS26 is not present in bacteria, but most archaea (see Results section).

Reviewer #2- Expertise: Solution NMR, Protein-Protein Interactions
(Remarks to the Author):

This is a solid paper describing the structure of the escortin Tsr2 alone and in complex with an intrinsically disordered peptide derived from the ribosomal protein Es26. The NMR work looks solid, and the ancillary biochemistry seems fine. The results are interesting and of general interest and I recommend acceptance.

Minor comment.

The right panel in Fig. 2b and Fig. 3b would look a lot clearer if protons were omitted from the structures.

We agree with the reviewer and have changed the figures accordingly.

REVIEWERS' COMMENTS:

Reviewer #1 (Remarks to the Author):

The modifications in the text and figures(especially the changes in the discussion) have greatly improved the manuscript. Now it is possible to appreciate the value of the work in the context of the field.

All my points have been addressed and the paper is now, in my opinion, acceptable for publication.